# Comparative Genome Analysis of 33 *Chlamydia* Strains Reveals Characteristic Features of *Chlamydia Psittaci* and Closely Related Species

**DOI:** 10.3390/pathogens9110899

**Published:** 2020-10-28

**Authors:** Martin Hölzer, Lisa-Marie Barf, Kevin Lamkiewicz, Fabien Vorimore, Marie Lataretu, Alison Favaroni, Christiane Schnee, Karine Laroucau, Manja Marz, Konrad Sachse

**Affiliations:** 1RNA Bioinformatics and High-Throughput Analysis, Friedrich-Schiller-Universität Jena, 07743 Jena, Germany; martin.hoelzer@uni-jena.de (M.H.); lisa-marie.barf@uni-jena.de (L.-M.B.); kevin.lamkiewicz@uni-jena.de (K.L.); marie.lataretu@uni-jena.de (M.L.); manja@uni-jena.de (M.M.); 2Animal Health Laboratory, Bacterial Zoonoses Unit, University Paris-Est, Anses, 94706 Maisons-Alfort, France; Fabien.Vorimore@anses.fr (F.V.); Karine.Laroucau@anses.fr (K.L.); 3Institute of Molecular Pathogenesis, Friedrich-Loeffler-Institut (Federal Research Institute for Animal Health), 07743 Jena, Germany; alison.favaroni@fli.de (A.F.); christiane.schnee@fli.de (C.S.)

**Keywords:** *Chlamydia*, *Chlamydia psittaci*, *Chlamydia trachomatis*, genome analysis, annotation tool, core genome, plasticity zone, variable genomic sites, host preference

## Abstract

To identify genome-based features characteristic of the avian and human pathogen *Chlamydia (C.) psittaci* and related chlamydiae, we analyzed whole-genome sequences of 33 strains belonging to 12 species. Using a novel genome analysis tool termed Roary ILP Bacterial Annotation Pipeline (RIBAP), this panel of strains was shown to share a large core genome comprising 784 genes and representing approximately 80% of individual genomes. Analyzing the most variable genomic sites, we identified a set of features of *C. psittaci* that in its entirety is characteristic of this species: (i) a relatively short plasticity zone of less than 30,000 nt without a tryptophan operon (also in *C. abortus, C. avium, C. gallinacea, C. pneumoniae*), (ii) a characteristic set of of Inc proteins comprising IncA, B, C, V, X, Y (with homologs in *C. abortus, C. caviae* and *C. felis* as closest relatives), (iii) a 502-aa SinC protein, the largest among *Chlamydia* spp., and (iv) an elevated number of Pmp proteins of subtype G (14 in *C. psittaci*, 14 in *Cand.* C. ibidis). In combination with future functional studies, the common and distinctive criteria revealed in this study provide important clues for understanding the complexity of host-specific behavior of individual *Chlamydia* spp.

## 1. Introduction

Chlamydiae are different from typical eubacteria for their obligate intracellular nature, which manifests itself in a biphasic developmental cycle comprising extracellular and intracellular stages. Along this cycle, infectious, but metabolically largely inactive elementary bodies (EBs) enter host cells to transform into non-infectious, metabolically active reticulate bodies (RBs) within a vacuole-like inclusion. These RBs replicate causing expansion of the inclusion and differentiate back into EBs to start a fresh cycle after host cell rupture.

The genomes of all *Chlamydia* spp. have undergone massive condensation in the course of co-evolution with eukaryotic host cells. In contrast to other parasitic and symbiotic microorganisms, this reduction resulted from genome streamlining rather than degradation [1].

The average chlamydial genome size of 1 Mbp is indeed lower than in typical eubacteria. A reduced genome implies the absence of loci encoding essential cellular pathways. Thus, chlamydiae rely on host cells for nutrients, such as amino acids, nucleotides and lipids [2,3] since they are incapable of synthesizing these substrates. Instead, they seem to compensate for this by co-opting suitable cellular pathways that provide the necessary nutrients [4,5]. On the other hand, recent genome analysis helped to qualify the long-held hypothesis of chlamydiae being ‘energy parasites’ [6] by revealing the presence of a metabolic chain leading to ATP production [7,8,9].

One of the factors complicating research on etiology and pathology of chlamydia infections is the low number of proven virulence factors compared to many other bacteria. Nevertheless, there are important pathogens among the currently accepted 11 chlamydial species. For instance, *Chlamydia (C.) trachomatis* is an important human pathogen infecting the urogenital tract and eyes causing sexually transmitted disease [10] or trachoma [11], respectively. *C. pneumoniae* affects the human respiratory tract being among the main causative agents of community-acquired pneumonia [12].

*C. psittaci* can be an economically relevant pathogen in poultry and pet birds, where it causes avian chlamydiosis, and also a human pathogen causing atypical pneumonia after zoonotic transmission [13]. The recent discovery of *C. gallinacea* and *C. avium*, which occur mainly in poultry or pigeons, respectively, has added two more members of the genus *Chlamydia* with a host preference for *Aves* [14,15,16,17]. Wildlife birds can be a reservoir of more exotic chlamydial species, such as *Candidatus* C. ibidis, which was found in an ibis [18], or *C. buteonis* in hawks [19].

*C. abortus* strains are endemic in small ruminants representing a cause of late-term abortion in sheep and goats [20], as well as zoonotic transmission. As a recent report on isolates from wild birds suggested, some strains of *C. abortus* seem to have an affinity to avian hosts as well [21].

Among the remaining *Chlamydia* spp., there are facultative pathogens of cattle (*C. pecorum*), swine (*C. suis*), guinea pigs (*C. caviae*), cats (*C. felis*) and mice (*C. muridarum*) [20,22].

The requirement for cell culture and other unique characteristics have been causing specific experimental challenges in chlamydia research. For instance, genetic manipulation was achieved later than for other bacteria [23], and so far only a few chlamydia laboratories have succeeded in implementing the technology. In addition, certain strains are difficult to grow in cell culture and adequate cell-free axenic media are not available.

In this context, analysis of whole-genome sequences is an efficient way to characterize strains of interest and provide clues predicting or explaining certain phenotypic traits.

As more and more *Chlamydia* spp. genome sequences became available in recent years, a number of comparative studies focusing on *C. trachomatis* [7,24], *C. pneumoniae* [25,26], *C. psittaci* [27,28,29], *C. abortus* [30] and others [31] were conducted. These studies typically focused on human chlamydial pathogens [32] or investigated genetic and evolutionary relationships within the order *Chlamydiales* [33]. The present study focused on characterization of chlamydial species with avian host preference and comparative analysis including all members of the Genus *Chlamydia*.

As revealed by comparative studies, the genomes of *Chlamydia* spp. share a conserved synteny, i.e., they are highly conserved in gene content and gene order and, consequently, also their metabolic capacities [9,33]. On the other hand, chlamydial species display significant differences in terms of tissue tropism, host preference, immune and stress response patterns, as well as pathogenicity. To identify and explain genome-based peculiarities at species and strain levels is, therefore, a central task of comparative genomics. This implies the analysis of lower-synteny genomic regions, such as the hyper-variable region near the predicted replication termination region known as the plasticity zone (PZ), which harbors the tryptophan (Trp) biosynthesis operon, an important distinctive feature among *Chlamydia* spp. [25,34,35].

In addition, the inclusion membrane (Inc) proteins form a large family whose members are inserted in the inclusion membrane via type III secretion. Being exposed to the cytosol, some of them are among the major immunogens [36]. It is remarkable that, in the average chlamydial genome, approximately 4 percent of the coding capacity is dedicated to this family [37]. Furthermore, all chlamydial species harbor polymorphic membrane proteins (Pmps), which represent autotransporters with surface-exposed and membrane-translocated domains. They are regarded as virulence factors [38], as well as adhesins and immune modulators [39]. Due to its central regulatory role in differentiation between EBs and RBs, the histone-like protein pair HctA/B [40] could be of interest in the context of strain viability and growth characteristics.

The present study was based on comprehensive comparative analysis that included 33 strains of all species of *Chlamydia* validly published by March 2020, irrespective of their status as a pathogen, co-infecting agent or commensal. Publicly available genome sequences were complemented by *de novo* sequenced and assembled genomes of eight field strains of avian chlamydiae (*C. avium, C. gallinacea* and *C. abortus*). We anticipated that exploration of inter-species genomic diversity throughout the genus *Chlamydia* could entail advances in revealing distinctive properties of the avian and human pathogen *C. psittaci* and other chlamydiae with avian host preference.

## 2. Results

### 2.1. General Characteristics of the Genome Sequences

Basic genomic parameters of all 33 strains are given in Table 1. While *C. pneumoniae* strain TW-183 has the largest genome of this panel (sized 1,225,935 bp), strains of C. avium (10DC88 1,041,170 bp) and *C. trachomatis* (434-Bu 1,038,842 bp) possess the smallest genomes.

Between 89 and 93% of individual genomes carry coding sequences (CDS). The total number of CDS ranges from 887 (*C. muridarum*) to 1050 (*C. pneumoniae*), but only 53 to 61% of them have been assigned a specific function by the annotation software. The remaining 39–47% are still at hypothetical protein status.

### 2.2. Common and Unique Elements in the Genomes of Chlamydia spp.

Each grouping of chlamydial organisms, e.g., at species or genus level, can be described by its pan-genome, which comprises all genes found in all strains included. It can be subdivided into the core genome containing genes present in all strains and a dispensable or accessory genome comprising genes encountered in a single or a few strains [41]. In this study, RIBAP, a specially designed pipeline was used to calculate both core and pan-genomes.

The pan-genome of the present panel of 33 chlamydial strains comprised a total of 31,573 CDS (Table 1) forming 11,054 homologous clusters consisting of gene homologs (Roary output at 95% identity, including clusters with a single CDS). Through pairwise ILP comparisons within RIBAP, these clusters were refined into 1583 RIBAP groups with 784 groups including genes from all 33 input genomes.

To visualize intersecting sets, i.e., shared genes among the chlamydial species, UpSet diagrams involving all 33 strains (Appendix A) as well as the 12 type and reference strains (Figure 1) have been constructed. While 791 genes are common to the 12-strain panel (compared to 784 common to the complete 33-strain panel), it is obvious that *C. trachomatis* and its closest relatives share comparatively few genes outside the core genome with *C. psittaci, C. abortus, C. avium* and *C. gallinacea*. For instance, there are as few as 13 genes that *C. trachomatis* shares with *C. abortus* and *C. psittaci* only (Figure 1, i.e., 7 genes in column 19 plus 6 in column 23). In contrast, the more closely related species of *C. psittaci, C. gallinacea, C. avium* and *C. abortus* have an additional 62 genes in common, while *C. psittaci* and *C. abortus* share 126 more genes outside the core genome. The *C. pneumoniae* genome, the largest in this dataset, possesses the highest number of unique genes (121), but also *Cand*. C. ibidis (41) and *C. avium* (49) with its relatively small genome have an elevated number of singletons.

The core genome of all 33 strains comprised 784 genes (Appendix A), i.e., they share about four-fifths of their genome.

Concatenated core genes of all strains were used to reconstruct a phylogenetic tree (Figure 2). Two major monophyletic groups can be distinguished, the first one containing *C. trachomatis, C. muridarum* and *C. suis*, hence referred to as ‘the trachomatis group’, and the second one harboring the remaining 9 taxa. In the latter, *C. pecorum* and *C. pneumoniae* make up their own clade, while the other clade contains 7 species. At the second node, *Candidatus* C. ibidis forms an external branch, whereas *C. avium/C. gallinacea, C. felis/C. caviae*, and *C. abortus/C. psittaci* appear in further clades, each containing two species. We will refer to the members of these three two-species clades as ‘the psittaci cluster’.

The tree also reveals particularly high genetic heterogeneity within the species of *C. psittaci* and *C. gallinacea*. Notably, the atypical *C. abortus* strain 16DC122, which was isolated from a duck, forms its own branch apart from two typical ruminant strains of the species.

### 2.3. The Plasticity Zone (PZ)

Major hotspots for variable regions are observed at ~300,000 bp and ~600,000 bp (normalized genome sequences with *hemB* gene in initial position). The first hotspot is the hypervariable region of the plasticity zone (PZ).

There is only a low degree of similarity among *Chlamydia* spp. as seen with nucleotide identity values of PZ sequences, which are typically in the range of 30–50% (Table 2, Appendix A). *C. psittaci* 6BC and *C. abortus* S26-3 share the most similar structure with 81.55% nucleotide identities, whereas the avian *C. abortus* strain 16DC122 has a less similar PZ compared to strain S26-3 (67.41% identity). Remarkably, the closely related species of *C. avium* and *C. gallinacea* lack any PZ homology.

Basic PZ parameters of all 33 strains are given in Appendix A, representative strains of each species are compiled in Table 3. *C. suis* (82,505 nt), *C. muridarum* (82,115 nt) and *C. trachomatis* (52149-56792 nt) were found to have the largest PZs, while strains of *C. avium* (4669-5694 nt), *C. pneumoniae* (8759 nt) and *C. gallinacea* (15,845–16,624 nt) harbor considerably reduced versions.

The composition of the PZ was found to differ widely among the species and, to a lesser extent, also within species. While biotin modification genes *accB* and *accC* were identified at one PZ boundary in all 33 strains, purine synthesis genes *guaA* and *guaB*, which are usually marking the other PZ boundary, were missing in *C. avium*, *C. gallinacea*, *C. suis* and *C. trachomatis*. A complete *guaAB*-ADA operon was seen in *C. caviae*, *C. felis*, *C. muridarum*, *C. pecorum* and most of the *C. psittaci* strains. Functional Trp operons, which include the regulatory region, structural genes *trpA, B, D, F, kynU* and repressor gene *trpR*, were encountered in *C. caviae* and *C. felis*. The analogous region in *C. trachomatis* and *C. suis* consisted of three genes only. All other *Chlamydia* spp. genomes lacked a Trp operon.

The gene encoding the large cytotoxin *(toxB)* was found in all species except *C. avium, C. pneumoniae* and *C. trachomatis*. Its size ranged from 7053 nt in *C. psittaci* strain VS225 to 10,317 nt in *C. pecorum*. In *C. suis* and *C. pecorum*, two gene copies were identified, in *C. muridarum* even three. In *C. psittaci*, sequence and size variations were observed from strain to strain (Appendix A). In the case of *C. abortus*, it is remarkable that a 9312-nt *toxB* gene was seen in the avian strain 16DC122, whereas it is absent in typical ruminant strains, such as S26-3 and C18-98 (B577).

Genes encoding membrane attack complex/perforin proteins (MACPF) were identified in *C. felis*, *Cand.* C. ibidis, *C. muridarum*, *C. pecorum*, *C. pneumoniae*, *C. psittaci*, *C. suis*, and *C. trachomatis*. Smaller CDS annotated as MAC/perforin domain-containing protein in *C. abortus* and *C. caviae* (also in *C. felis* and *C. psittaci*) are probably pseudogenes.

### 2.4. Genes Encoding Polymorphic Membrane Proteins (pmps)

The above-mentioned hotspot regions also harbor clusters of *pmp* genes. To identify homologs and characterize the spectrum of *pmp* genes present in strains of the ‘psittaci cluster’, all previously annotated *pmp* sequences of *C. psittaci, C. avium, C. gallinacea* and *C. abortus* were blasted against the genome sequences of all 33 strains. While 21 individual *pmp* genes were seen in *C. psittaci* and 18 in *C. abortus*, their number was considerably reduced in *C. gallinacea* (10) and *C. avium* (7). The results in Table 4 also show that all known Pmp subtypes A, B, D, E, G, and H are represented by at least one member in each species. Subtype G proved the most extensive one with 11 members in *C. abortus*, 14 in *C. psittaci*, four in *C. gallinacea* and two in *C. avium*.

Appendix A illustrates the extent of sequence variation among strains of these four species. Notably, the set of individual *pmp* genes was conserved within a given species, where strain-to-strain variations can cover similarity values between 50 and 80%. The more conserved Pmp family members, such as PmpD, can be used for phylogenetic purposes. While sequence similarity within the same Pmp subtype among *Chlamydia* spp. is generally low, phylogenetic analysis revealed a genetic relatedness closely resembling genome-based data shown in Figure 1 and Appendix A. Comparison of PmpD proteins revealed higher than 50% similarity at amino acid level among the species of the ‘psittaci cluster’, but lower similarity to the ‘trachomatis group’ (Appendix A).

### 2.5. Inc Proteins

In contrast to the ‘trachomatis group’, where at least eight Inc family subtypes, i.e., IncA-G and V, and more individual members can be found, the remaining *Chlamydia* spp. possess less *inc* genes (Table 5). Six different *inc* gene types have been identified in *C. psittaci* and *C. abortus*, but two of them, incX and incY (or NC), have yet to be assigned to a particular subtype. Again, *C. avium* and *C. gallinacea* appear to have undergone a reduction of this locus, as they presented only subtypes A, B, C and V.

The Inc proteins are highly variable among the species of *Chlamydia*. As an example, IncB of the closely related species of *C. psittaci* and *C. abortus* share only 74.26% amino acid sequence similarity (see Appendix A). At the same time, Inc sequences are remarkably conserved within species. For instance, Inc sequence identities within *C. psittaci* and *C. abortus* are typically close to 100%, within *C. avium* and *C. gallinacea* higher than 90%.

### 2.6. The Secreted Inner Nuclear Membrane-Associated Chlamydia Protein (SINC)

Homologs of the gene encoding SINC of *C. psittaci* 6BC were identified in all ten strains of this species, with amino acid similarities between 86.2% (strains GR9 and WS-RT-E30), 96.8% (NJ1) and 99–100% (rest of the strains). SINC orthologs of reduced size were encountered in *C. abortus, C. caviae, C. felis, C. gallinacea, Cand*. C. ibidis, and *C. avium* (order of descending amino acid similarity, see Table 6 and Appendix A).

### 2.7. Histone-Like Proteins HctA and HctB

This protein pair was present in most of the *Chlamydia* spp. However, *C. avium* and *C. gallinacea* strains completely lacked the *hctA* gene (Table 6). In addition, *hctB* was missing in *C. gallinacea* strain JX-1 while present in the other field strains. In *C. avium*, only type strain 10DC88 harbors a truncated *hctB* gene sized 258 nt or 86 aa (probably a pseudogene) that was highly homologous to its counterpart in *C. gallinacea*. In the other *C. avium* strains, the *hctA/B* pair seems to be absent altogether.

### 2.8. Pseudogenes

To check genomes for the presence of homologs to *C. trachomatis* pseudogenes we conducted a blastn 2.7.1+ search with default parameters using known pseudogenes of *C. trachomatis* (*n* = 15) as queries. The latter were extracted from [42]. We found that two of the pseudogenes tested were conserved in all 33 genomes (CTL0228 and CTL0627 with 70–83% sequence identity and 76–86% sequence identity, respectively). These are the only homologous pseudogenes found in strains of *C. psittaci, C. avium, C. gallinacea C. abortus* and *C. pneumoniae*. As expected, more of the *C. trachomatis* pseudogenes are shared with the phylogenetically closer relatives of *C. suis* (9) and *C. muridarum* (7), but also *C. pecorum* (4) and *Cand*. C. ibidis (3) harbor additional homologs (Appendix A).

## 3. Discussion

Many bacterial genome sequences in public databases are in a permanent draft state [43]. Failure in achieving complete genome sequence assembly is often due to multiple sequence repeats in certain loci, which may confound assembly tools. However, incompletely assembled genome sequences pose limitations on accurate annotation and subsequent genome analysis [44]. In the present study, we used 27 completely and 6 nearly completely assembled genomes of chlamydial strains from 12 species, 25 of which had already been in the public domain. In addition, 4 strains of *C. gallinacea,* 3 of *C. avium* and 1 of *C. abortus* were *de novo* sequenced and assembled.

### 3.1. Bioinformatics Tools: New and Unique Features of the RIBAP

While several tools for pan-genome calculation were already available (such as Roary), we observed that the core genome size was often underestimated, in particular, when evolutionarily distinct species were compared. Thus, in the course of this comparative study, we developed a combined approach by connecting initial Roary groups based on high sequence similarity (95%) with additional parameters derived from evolutionary pairwise ILP calculations (github.com/hoelzer-lab/ribap). Using this approach, we are able to calculate a comprehensive core genome, even at genus level and for evolutionarily distinct bacteria.

In addition, the RIBAP code is public domain and the pipeline is implemented using a workflow management system with containerized steps, thus allowing easy and reproducible execution on different platforms and modular pipeline adjustments including versionization in the future.

One output of RIBAP is a concise HTML table (see osf.io/j9zas for download or direct HTML access) that compiles not only the core genome, but also other gene groups belonging to the accessory genome. The table allows an interactive search for specific genes of interest and visualizes sequence similarity thresholds and phylogeny for every single gene group. When the number of input genomes to be compared becomes too large for visualization in a table format, an UpSet plot as shown in Figure 1 (and Appendix A) will still allow quantitative visualization of multiple gene set intersections. An example of the RIBAP output concerning the HctB protein is shown in Figure 3.

Importantly, our approach also encompasses genes that lack a functional annotation due to assembly errors, annotation problems or incomplete reference databases (so-called ‘hypothetical genes’). Thus, we are able to determine a more complete set of core genes and even assign functions *a posteriori*. Hypothetical genes can be added to a RIBAP group consisting of genes with an assigned function, thus allowing annotation of genes that had been overlooked by the annotation software. For example, the RIBAP group13 (see HTML table located at https://osf.io/j9zas/) includes genes present in all 33 strains and functionally annotated as *rpmG*, which encodes the 50S ribosomal protein L33. However, for *C. trachomatis* 434-Bu, Prokka only reported a hypothetical gene despite clear sequence similarity to the other *rpmG* genes in this group. By exclusion or routine handling of ’hypothetical genes’, this group would have lacked one of the 33 strains and thus would not have been included in the core gene set. Furthermore, our analysis also revealed additional core genes (annotated as ‘hypothetical proteins’) present in all 33 strains, which are interesting targets for further studies and additional annotation efforts.

### 3.2. Core Genome vs. Dispensable Genome

As a central parameter in comparative genomic studies, the core genome comprises the genes or CDS shared by all strains of the selected panel. However, there is no general agreement on definition of the term itself, nor on similarity cut-off values to be applied.

In this situation, core genome size naturally depends on the selection of strains studied and the calculation method including sequence similarity thresholds.

As standard Venn and Euler diagrams are an inadequate solution for quantitative visualization of multiple (n > 4) gene set intersections, we used the UpSetR package (Figure 1, Appendix A), a scalable alternative for visualizing intersecting sets and their properties [45]. The result of 784 genes shared among 33 chlamydial strains (or 791 among 12 type strains) is somewhat higher than the number reported in the study by Joseph et al. [32], who identified 668 core genes in a panel of 36 strains of *Chlamydiaceae*. These authors studied a different panel of strains with emphasis on *C. pneumoniae, C. pecorum* and *C. psittaci/C. abortus* and defined core genes as protein-coding gene clusters shared by all strains. Sigalova et al. identified 698 orthologous groups (genes) that were universally conserved in a huge set of 227 genomes of 16 species and *Candidati* of *Chlamydia* [1].

Investigating genome synteny in higher taxonomic hierarchies, Collingro et al. [33] previously found 560 genes belonging to the core genome of the phylum *Chlamydiae*, but nine years later, based on a far larger number of organisms and genome sequences, that number decreased to 340 [46]. In another study, Pillonel et al. [47] identified 424 core proteins shared by members of the order *Chlamydiales*.

The higher number of core genes in our study can be explained by our combined approach: We relied on the same annotation source for all genomes (re-annotation with Prokka) and also included all ‘hypothetical proteins’ (39–46% of all CDS in our annotations) in the comparison. This approach became possible because we based our study on gene homology rather than gene function. By including non-annotated (‘hypothetical’) genes, often omitted in other studies, we provide a more comprehensive comparative genome analysis.

In addition, our novel ILP approach is able to connect even gene groups that share low sequence similarity (Figure 3), potentially resulting in additional core gene sets that are missed by other approaches. It should be emphasized that membership of the core genome is based on a sequence identity threshold of 95% within Roary clusters, which are further connected by supporting ILP results based on an all-vs-all MMSeqs2 comparison without any sequence identity cutoff. Thus, the combination of re-annotation of all genomes based on the same database and novel functionalities of RIBAP accounts for the larger set of core genes identified in our study.

Using the new RIBAP pipeline, we selected the following criteria and thresholds for core genome identification: We define a protein-coding gene to be part of the core genome when it forms a RIBAP group of 33 members (i.e., the number of genomes included in this study). Thereby, a RIBAP group consists of at least one Roary cluster with a sequence identity of 95%. Multiple 95% Roary clusters can be connected via ILP support into larger RIBAP groups, potentially forming a core gene. We believe that these conditions take the peculiarities of chlamydial genomes into account and may represent a working compromise for core genome definition.

In an alternative approach, van Aggelen et al. [48] recently proposed a core genome definition to be based on conserved sequences rather than conserved genes. In these circumstances, larger core genome sizes will be calculated, since also non-coding sequences are included.

The elements forming the core genome are considered to be indispensable and represent an accurate dataset for phylogenetic studies [49,50]. Therefore, we reconstructed the phylogenetic tree of *Chlamydia* spp. using concatenated core gene sequences of the strains examined (Figure 2). While largely confirming previously published phylogenies based on rRNA [51], conserved proteins [47] and core locally collinear blocks [32], this tree comprises all validly published species of the genus *Chlamydia* and highlights intra-species genetic variability of *C. psittaci* and *C. gallinacea*. It is worth noting that all chlamydial organisms originating from avian hosts have been included in ‘the psittaci cluster’ following the second node, thus confirming their close evolutionary relationship. This is in line with our observation that the four species of *C. psittaci, C. gallinacea, C. avium* and also *C. abortus* share another 68 common genes in addition to those of the core genome (Figure 1). *C. abortus, C. caviae* and *C. felis*, which have a preference for non-avian hosts, separated from the common ancestor at a later stage, so that they are still closely related.

### 3.3. The Plasticity Zone

The data of this study show that the absence of a tryptophan (Trp) operon is a common feature of the avian species *C. psittaci, C. gallinacea, C. avium,* and the closely related *C. abortus*. The fact that *C. abortus* with its mainly ruminant and only occasional avian strains also lacks the operon is not surprising as it evolved from *C. psittaci* relatively recently [30,32].

Functional operons, which include the regulatory region, structural genes *trpA, B, C, D, F, kynU* and repressor gene *trpR*, were encountered in *C. caviae* and *C. felis*. In the case of *C. trachomatis*, it was suggested that genital strains possess the operon, while ocular strains do not [52]. Our analysis revealed *trpA, trpB* and *trpR* genes in all three *C. trachomatis* strains, but the ocular strain A-Har-13 had two shortened CDS encoding *trpB* segments instead of a single *trpB* gene, which may indicate its limited functionality. In *C. muridarum*, we found no *trp* genes, thus confirming the absence of a functional operon, although the presence of truncated remnants as suggested by Xie et al. [52] cannot be ruled out. Similarly, *C. pecorum* type strain E58 lacked *trp* genes in the PZ, but a locus containing six *trp* genes outside the PZ was recently reported [1]. Besides the absence of a Trp operon, chlamydial strains of *C. psittaci*, *C. abortus, C. avium, C. gallinacea,* as well as *C. pneumoniae* share a tendency towards reduced PZ size (Table 4).

Regarding the most prominent locus in the PZ, the role of the large cytotoxin as a potential virulence factor is far from understood. *ToxB* was found in all species except *C. avium, C. pneumoniae* and *C. trachomatis*. In the present analysis, the *toxB* gene product has been categorized as an ortholog of lymphostatin/EFA-1, a toxin from *E. coli* (EPEC and EHEC) that also occurs in *Citrobacter rodentium* besides chlamydiae. It harbors three enzymatic activities associated with glycosyltransferase – (D-X-D, 1.6 kb), protease–(C, H, D, 4.5, 4.8 kb), and aminotransferase (TMGKALSASA, 5.8 kb) motifs [53]. While in *E. coli* the *lifA/efa-1* gene is present in pathogenic, but absent in non-pathogenic strains, there is no analogous data for chlamydiae. Based on similarity to the cytotoxin of *Clostridoides difficile*, the glycosyltransferase domain was suggested to interact with eukaryotic host cells by glycosylating proteins of the Ras superfamily, thus inactivating them and causing disassembly of the actin cytoskeleton [54,55]. Other studies highlighted homologies of the chlamydial toxin to clostridial large cytotoxins and yersinial type III effector YopT, which are involved in protein translocation and disassembly of the host cytoskeleton [56,57].

Functional members of the MAC/perforin family were encountered in most of the species considered, but not in *C. avium* and *C. gallinacea*, nor in *C. abortus* and *C. caviae*, where truncated genes or pseudogenes were found. However, these findings remain provisional as the accuracy of annotation tools is limited in this regard. All MAC/perforin proteins possess a MACPF domain, but the degree of conservation within the family is low, so that sequence-based search algorithms may not always be suitable for identification. Their assumed function consists in pore formation, which may contribute to host cell entry [58].

With so many question marks left, there is a notion that PZ genes, even if translated into functional proteins, may be dispensable for infectious processes [59]. Nevertheless, the present finding of *toxB* in an atypical *C. abortus* strain from an avian source, which contrasts the absence of the cytotoxin gene in typical ruminant *C. abortus*, may be an indication of a role in host tropism, even though it is also missing in the *C. avium* genome. All in all, it has yet to be experimentally shown that this toxin can be a specific virulence factor in *C. psittaci* and closely related avian chlamydiae.

There is also a notable diversity among *Chlamydia* spp. regarding the *guaAB*/ADA locus. While *C. caviae, C. felis, C. pecorum, C. muridarum* and most of the *C. psittaci* strains harbor an intact operon it is absent in the other species (Table 3). As the operon’s gene products are involved in salvaging biosynthesis of purine nucleotides, which are required for bacterial growth, this indicates different purine salvage strategies [60]. In the case of *C. psittaci*, the two strains lacking *guaAB*/ADA, GR9 and WS-RT-E30, form their own clade in the phylogenetic tree in Figure 2. The presence of the *guaAB*/ADA operon was suggested to reflect adaption of species or strains to a niche with biochemical restrictions [29]. However the distribution of this trait among species and strains in our study did not reflect phylogeny or host tropism, so that a role of the operon as a major delineator of speciation cannot be anticipated.

### 3.4. The Family of Polymorphic Membrane Proteins (Pmps)

The Pmps represent autotransporters carrying conserved tetrapeptide motifs in the central part or passenger domain. While sharing many of the characteristics of classical autotransporters in eubacteria [38], Pmps are unique to chlamydiae. Some of them were shown to be adhesins and immune modulators in *C. pneumoniae* and *C. trachomatis* [39,61]. Since *Chlamydia* spp. use a high proportion of their genome to encode this protein family (*C. pneumoniae* 17.5%; *C. trachomatis* 13.6%) it is likely that they are of crucial importance.

An unusually high number of mutated sites in these loci is responsible for high sequence variation across species [62,63]. The data of our study highlight the extreme interspecies diversity among members of this gene/protein family. At the same time, the spectrum of *pmp* genes proved fairly stable within the same species (Table 5 and Appendix A). Due to low inter-species sequence homologies of individual Pmp family members, annotation can be complicated. In the case of hitherto unassigned hypothetical proteins, the software would only recognize sufficiently similar sequences as homologs of known proteins, which could prevent identification of certain orthologs.

In this study, all species were found to harbor the complete set of Pmp subtypes A, B, D, E, G, and H. In the *C. avium* strains, a reduced set consisting of only 7 members was identified, among them only two subtype G proteins. Most species of the ‘psittaci cluster’ are distinguished from the trachomatis group (with 9 Pmps) by a larger number of individual Pmps ranging from 21 in *C. psittaci* to 10 in *C. gallinacea*. The species of *C. psittaci* and *C. gallinacea* display the highest intra-species sequence variation in the Pmp family (Appendix A).

With the exception of *C. avium*, the ‘psittaci cluster’ contains an elevated number (≥4) of subtype G members, in contrast to the trachomatis group, which has only two members, PmpG and PmpI. In the case of *C. psittaci*, other workers had already noted the presence of several *pmpG* genes [35] and their variability among genotypes [64]. The present data show that the *pmpG* subtype of this pathogen exhibits the greatest variety among all *Chlamydia* spp., with 14 currently known members (*pmp*7–17 and *pmp*19–21).

Considering our current knowledge on Pmps, which includes experimentally proven adhesive functions [39,61] and immunogenicity [65,66] it is straightforward to suggest a substantial role in host tropism and adaptation, as well as species-specific pathogenicity [67].

### 3.5. Inclusion Membrane Proteins

As the presence of a bi-lobed hydrophobic domain of 40–60 amino acids is the only common feature, and inter-species similarity of primary sequences is low, it is difficult to identify all members of the Inc protein family solely using bioinformatics tools. In *C. trachomatis*, 39–59 Inc proteins have been predicted, but only 23 of them functionally characterized [68]. Nevertheless, due to being located in the inclusion membrane, these proteins are players in direct interaction with host defense factors, which suggests a major role in pathogenesis. In addition to the known subtypes A-G, a new family member IncV was recently identified and shown to be directly involved in the formation of membrane contact sites with host cells [69]. The fact that Incs are among the major immunogens [36,70] confirms their active participation in pathogen-host interaction and also recommends them as efficient species-specific capture antigens in serology and potential vaccine components. In the present study, we have identified a tendency towards reduction in Inc family size among some avian *Chlamydia* spp., particularly in *C. gallinacea* and *C. avium*. This could be a contributing element to their particular host preference. It is also obvious that IncB sequences of ‘psittaci cluster’ members are more similar to each other than to the remaining *Chlamydia* spp. (Appendix A).

### 3.6. SINC Protein

This type III-secreted effector is considered a potential virulence factor for its ability to target the nuclear envelope [71]. We have identified the *sinC* locus encoding a 502/503-aa protein in all ten strains of *C. psittaci*, which is by far the largest SinC among *Chlamydia* spp.

In strain Mat116 we found a truncated version consisting of two overlapping sequence strands. In the other species of the ‘psittaci cluster’, there were SINC orthologs shortened to roughly 50–70% of the original size (Table 6). Mojica et al. were able to show that the shorter SINC orthologs in the closely related species of *C. abortus* and *C. caviae* still retained their specific functionality [71]. On the other hand, they found a ‘weak ortholog’ in *C. trachomatis*, which has only 12.5% sequence similarity and lacks the specific capability of nuclear membrane association. Our own multiple Blast query did not identify SINC orthologs outside the ‘psittaci cluster’. This could indicate a role of this protein in host tropism and pathogenesis.

### 3.7. Histone-Like Proteins HctA and HctB

Although the *hct* locus is *a priori* not considered a particularly variable genomic site there is a striking contrast between high intra-species conservation and low inter-species homology of these proteins. For instance, sizeable homology of HctB proteins is only observed between *C. psittaci* and *C. abortus* (71–83% aa identity) as well as between the Hct-carrying *C. gallinacea* and *C. avium* strains (83–88%). Nevertheless, comparison among species reveals that HctBs within the ‘psittaci cluster’ are more similar to each other than to other chlamydial species (Appendix A).

This protein pair is known for its central role in RB-to-EB differentiation late in the developmental cycle [72,73], which includes the capability of repressing transcription and translation through binding to the genome. It is conceivable that sequence variations in this pair of lysine-rich and highly basic proteins could have implications on host preference and the ability to proliferate in host tissue. In this context, it is remarkable that *C. gallinacea* and *C. avium*, with their narrow host range and challenging cell culture, are obviously poorly equipped in terms of Hcts. In future research, a closer look at primary and secondary structures of chlamydial histone-like proteins combined with cell culture experiments including *hct* mutants could provide some answers to those questions.

### 3.8. Pseudogenes

Detailed studies on pseudogenes in chlamydiae are still scarce, but their number is believed to be low compared to other bacteria [1]. Although the issue has been addressed only superficially here, our findings suggest that pseudogenes encountered in *C. trachomatis* are not as prominent in phylogenetically distant chlamydial species. The latter, which include all *Chlamydia* spp. with avian host preference, probably have a separate set of pseudogenes. This in itself is an interesting point since it is conceivable that pseudogenes could be part of a marker set of host or tissue tropism. In any case, more systematic studies are needed to investigate this subject.

## 4. Materials and Methods

### 4.1. Chlamydial Strains

The 33 strains included in this study and the sources of whole-genome sequences are given in Table 6.

### 4.2. Genome Sequencing and Genome Assembly

The genomes of eight field strains (three *C. avium*, four *C. gallinacea*, one *C. abortus*), were *de novo* sequenced and assembled (Table 1). These strains were cultured on BGM cells and DNA extracted as described previously [14]. Approximately 5 µg of genomic DNA was sent to GATC/Eurofins Genomics (Konstanz, Germany) for Illumina MiSeq 2 × 300-bp paired-end sequencing (except *C. abortus* 16DC122 with 2 × 151 cycles). In the case of *C. gallinacea* strain 12-4358, a genomic library was prepared using 1 ng of genomic DNA and the Nextera XT kit (Illumina, Berlin, Germany, and sequencing was performed on the MiSeq platform at Anses using a Micro V2 with 2 × 151 cycles. Raw sequencing data were quality controlled using FASTQC [74]. Subsequently, the raw reads were assembled using SPAdes v3.12 [75] with k-mer values of 21, 33, 55, 77, 99, and 127, the careful option and automatic coverage cut-off. The assembly quality was evaluated using QUAST [74]. The raw sequencing data are deposited in the European Nucleotide Archive under accession PRJEB40883 and the final assemblies are available at https://osf.io/j9zas/.

### 4.3. Pan-Genome and Core Genome Calculation Using RIBAP

The Roary ILP Bacterial Annotation Pipeline (RIBAP) is a newly developed pipeline freely available at github.com/hoelzer-lab/ribap and implemented using the workflow management system Nextflow [76]. For an input set of genome sequences, the pipeline performs annotation, core gene set calculation of the identified coding genes, alignments and phylogenetic reconstructions of homologous genes and the full core gene set, as well as various visualizations of the results. Each step of the pipeline is encapsulated in its own Docker container, thus, only Nextflow and Docker are required to run RIBAP. For the present study, we ran the pipeline in v0.4 using the following command: nextflow run hoelzer-lab/ribap -r v0.4 --fasta ‘*.fasta’ --tree --tmlim 240.

We executed the pipeline twice: (1) on the full set of all 33 genomes and (2) only on the type strains of the 12 species included in this study. RIBAP will be described in full detail in an upcoming publication. Briefly, the pipeline combines the output of various tools, most importantly Roary and an ILP-approach, to calculate a comprehensive core gene set even in the case of evolutionarily different species. Below, we explain the core features of RIBAP to calculate a core gene set. Version numbers of all third-party tools involved in the execution of RIBAP are provided in version 04 of the GitHub release at github.com/hoelzer-lab/ribap. Complete results of the pipeline are provided at the Open Science Framework (osf.io/j9zas).

### 4.4. Annotation

First, to ensure full comparability of annotations using the same data basis, RIBAP re-annotates all 33 genome sequences using Prokka v1.14.5 [77] with default parameters. Prokka searches each candidate CDS (from start to stop codon) against a reference protein database derived from UniProtKB, while CDS with no database match are designated ‘hypothetical protein’. Since Prokka is an integral part of RIBAP, the subsequent core genome calculation is based on Prokka’s CDS annotations, which also include hypothetical proteins.

### 4.5. Pan- Genome Scaffold

Next, we calculated a preliminary scaffold of the core gene set using the rapid large-scale prokaryote pan-genome analysis tool Roary v3.13.0 [78] based on the Prokka CDS annotations. RIBAP automatically performs multiple Roary calculations with different sequence similarity thresholds (60%, 70%, 80%, 90%, 95%). However, the final RIBAP core gene set calculation is based on the 95% Roary output, whereas results of lower similarity thresholds are only used for visualization of the final core gene groups. The output consists in so-called Roary clusters, which are further connected into larger RIBAP groups using Integer Linear Programming (ILP) results.

### 4.6. Integer Linear Programming and GLPK

A disadvantage of pan-genomes produced by Roary at high sequence similarity thresholds is the fragmentation of clusters that likely belong together. On the other hand, a low sequence similarity threshold can result in an increased amount of false positive assignments (see Roary online FAQ). Therefore, within RIBAP, we adapted and extended an ILP approach [79] by an InDel-model to minimize the evolutionary distance between CDS based on an MMseqs2 (v10.6d92c) all-vs-all CDS comparison [80]. We split the initial MMseqs2 all-vs-all homology table into all pairwise genome comparisons and formatted the output to be solved with ILPs using the GNU Linear Programming Kit (GLPK, v4.65) package [81]. Subsequently, the solved ILPs were combined again and the resulting data was used to connect fragmented Roary clusters even below the sequence similarity threshold of 95%. We run RIBAP with the ‘--tmlim‘ parameter set to 240 s to limit how long GLPK will try to solve each individual ILP.

### 4.7. Creating a RIBAP Group

A RIBAP group (set of homologous genes) is created by merging the preliminary Roary pan-genome clusters and the refined pairwise ILP results. Using pairwise ILP data, smaller Roary clusters of low sequence similarity were connected into larger RIBAP groups that potentially represent homologous core genes. For detailed documentation refer to the scripts provided at https://github.com/hoelzer-lab/ribap.

### 4.8. The RIBAP Output

All results are compiled in a searchable and interactive HTML table embedding the final RIBAP groups and including gene designation, gene description, a color-coded heat map based on Roary assignments at different thresholds, as well as a phylogenetic tree based on the multiple sequence alignment (MSA) of the CDS making up a RIBAP group.

### 4.9. Phylogenetic Tree Based on Core Genomes

RIBAP further calculates a phylogenetic tree based on the core genome, comprising CDS present in all input genomes. The pipeline utilizes MAFFT v7.455 [82] to create an MSA for each RIBAP group and concatenates the resulting alignments. Then, RAxML v.8.2.12 [83] is applied to construct a phylogenetic maximum likelihood tree using a bootstrap value of 100 and the PROTGAMMAWAG model.

### 4.10. UpSet Diagrams

Standard Venn and Euler diagrams are an inadequate solution for quantitative visualization of multiple (*n* > 4) gene set intersections. Thus, we used the UpSetR package v1.4.0 in RIBAP, a scalable alternative method for visualizing intersecting sets and their properties [45]. As input for UpSetR, we selected all 33 genomes as well as type strains of the 12 species included in this study and their homologous gene groups identified with RIBAP. We restricted the visualization to the largest 40 intersecting sets.

### 4.11. Multiple Blast to Identify Homologs of Pmp, Inc and SinC Genes

All sequences of *pmp*/Pmp and *inc*/Inc family members of *C. psittaci, C. avium, C. gallinacea* and *C. abortus* type strains annotated in the NCBI (nucleotides) and/or UniProt (proteins) databases were compiled in multi-FASTA files (one query file per species) and blasted against the genome sequences of all 33 strains of this study. The resulting hits were sorted according to target strains and filtered at 50% sequence identity. Likewise, the amino acid sequence of SinC of *C. psittaci* (EGF85279.1) was blasted against all 33 genomes.

### 4.12. Calculation of Sequence Identities

Nucleotide and amino acid sequence identity values were calculated using distance matrices based on multiple sequence alignments in Geneious v. 10.2.4 (Biomatters Ltd., Auckland, New Zealand).

### 4.13. Normalization of Genomes

To facilitate visual genome comparison using generic tools, such Geneious, whole-genome sequences were rearranged by means of a custom python script (located as helper script in the RIBAP repository) to have the *hemB* (delta-aminolevulinic acid dehydratase) gene in the initial position.

## 5. Conclusions

Using a newly developed combined genome analysis approach we were able to accurately calculate the core genome of 33 strains of *Chlamydia* spp. belonging to 12 species, which comprises four-fifths of the respective genomes. Our analysis also revealed a high degree of genetic variability among field strains of the species of *C. psittaci* and *C. gallinacea*. An atypical *C. abortus* strain isolated from a bird was shown to share common traits with *Chlamydia* spp. of avian host preference.

Our study has revealed a number of characteristic features, which, in its entirety, distinguish *C. psittaci* from other *Chlamydia* spp.: (i) a relatively short PZ without a tryptophan operon, (ii) a characteristic set of Inc proteins (2–5), (iii) the presence of the largest *sinC* locus, and (iv) an elevated number of subtype G Pmp proteins. The fact that characteristic genomic traits common to all avian *Chlamydia* spp. could not be identified indicates that the clue for understanding chlamydial host preference may rather lie at the level of SNPs or small indel events as well as gene regulation.

Future studies should investigate the functions of these characteristic loci in more detail and address the implications for pathogen-host interactions.

## 6. Availability of Data and Materials

The datasets generated and/or analyzed during the current study are available in an OSF repository at https://osf.io/j9zas/. All code is available at https://github.com/hoelzer-lab/ribap. Novel sequencing data were uploaded to ENA acc.no. PRJEB40883 and assemblies generated for eight strains in the course of this study are available at https://osf.io/j9zas/.

## Figures and Tables

**Figure 1 pathogens-09-00899-f001:**
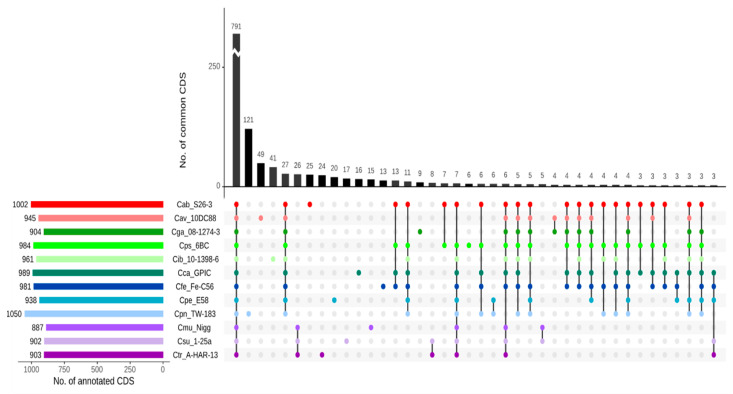
UpSet diagram illustrating the core genome and common genes among all 12 chlamydial species based on complete genome sequences of the type strains. The intersections are based on the RIBAP output for the 12 reference strains of *Chlamydia* species.

**Figure 2 pathogens-09-00899-f002:**
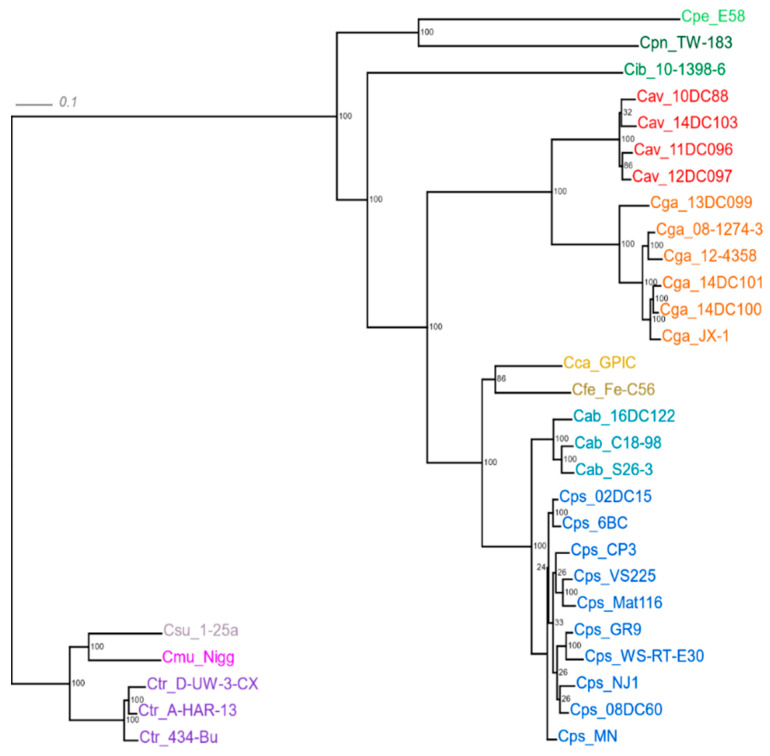
Phylogenetic reconstruction based on the concatenated multiple-sequence alignment (MSA) of the core genome predicted by RIBAP for all 33 genomes. The tree was constructed using RAxML maximum-likelihood algorithm with 100 bootstrap replicates. The scale above indicates 1% sequence divergence. Strains of the same species appear in the same color.

**Figure 3 pathogens-09-00899-f003:**
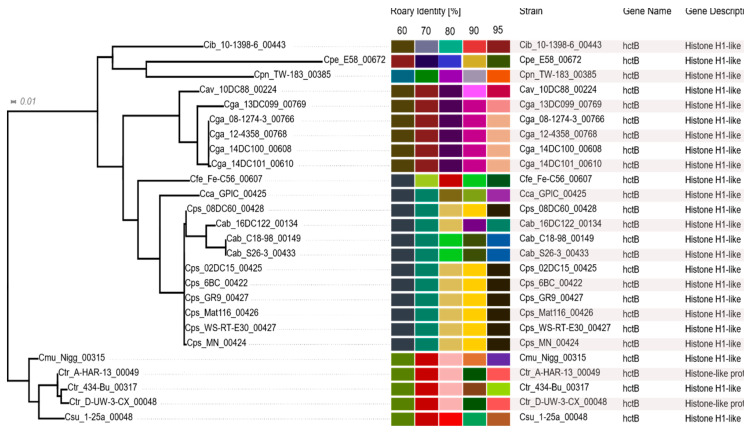
Polished screenshot of the RIBAP table displaying hctB genes encoding histone H1-like protein Hc2 (group 787) in 26 strains, their homology levels (color table) and a dendrogram based on multi-sequence alignment and FastTree. Displayed for each RIBAP group are the group size and all involved strains, as well as gene name and gene description for each strain. The corresponding assignment to Roary runs with sequence identities of 60, 70, 80, 90 and 95% is color coded in their respective groups. On the left-hand side, a phylogenetic tree is displayed, which is generated from the MSA of the CDS within this RIBAP group. Alignment and tree are provided for download. The full HTML table can be found in the OSF repository.

**Table 1 pathogens-09-00899-t001:** General parameters of genome sequences analyzed in this study.

Species_Strain	Genome Size [bp]	No. of Contigs ^a^	No. of CDS (Prokka)	Hypothetical Proteins [%]	Coding Density	Max. Protein Length [aa]	Min. Protein Length [aa]	Signal Peptides	G+C [%]	rRNA Operons	tRNA	tmRNA	misc_RNA
Cab_16DC122	1,131,589	1	992	44.15	0.89	3104	35	54	39.64	1	39	1	5
Cab_C18-98	1,161,363	4	1017	44.14	0.89	1807	36	66	39.90	1	43	1	6
Cab_S26-3	1,144,377	1	1002	43.51	0.90	1807	40	67	39.87	1	39	1	5
Cav_10DC88	1,041,170	1	945	42.53	0.91	1780	32	43	36.92	1	39	1	6
Cav_11DC096	1,040,930	1	902	41.35	0.90	1807	34	45	36.89	1	39	1	6
Cav_12DC097	1,041,858	1	904	41.59	0.89	1807	34	45	36.89	1	39	1	6
Cav_14DC103	1,042,060	1	907	41.45	0.89	1807	32	45	36.91	1	39	1	6
Cca_GPIC	1,173,390	1	989	42.87	0.91	3347	44	68	39.22	1	38	1	6
Cfe_Fe-C56	1,166,239	1	981	41.99	0.91	3299	36	62	39.38	1	38	1	6
Cga_08-1274-3	1,059,583	1	904	41.15	0.91	3121	35	43	37.94	1	39	1	7
Cga_12-4358	1,058,551	1	905	41.21	0.90	3251	35	44	37.94	1	39	1	7
Cga_13DC099	1,051,382	4	902	41.10	0.89	3257	46	38	37.83	1	39	1	7
Cga_14DC100	1,051,382	3	902	41.35	0.89	3251	35	41	37.92	1	39	1	7
Cga_14DC101	1,056,703	3	903	41.41	0.89	3251	35	40	37.92	1	39	1	7
Cga_JX-1	1,059,522	1	918	41.17	0.91	2648	35	44	37.93	1	39	1	7
Cib_10-1398-6	1,146,066	4	961	42.35	0.91	3126	31	60	38.32	1	38	1	5
Cmu_Nigg	1,072,950	1	887	39.12	0.90	3336	36	49	40.34	2	37	1	6
Cpe_E58	1,106,197	1	938	40.19	0.93	3439	40	53	41.08	1	39	1	5
Cpn_TW-183	1,225,935	1	1050	46.76	0.90	1827	32	60	40.58	1	38	1	5
Cps_02DC15	1,172,182	1	991	42.78	0.91	3078	43	66	39.06	1	39	1	5
Cps_08DC60	1,171,660	1	998	42.68	0.90	3255	43	67	39.05	1	39	1	5
Cps_6BC	1,172,032	1	984	42.78	0.91	3358	43	66	39.06	1	39	1	5
Cps_CP3	1,168,150	1	1062	44.53	0.90	3131	35	60	39.06	1	39	1	5
Cps_GR9	1,147,152	1	994	43.36	0.90	3104	39	60	39.08	1	39	1	5
Cps_Mat116	1,163,362	1	1003	43.96	0.89	3165	35	57	39.06	1	39	0	5
Cps_MN	1,168,490	1	1001	43.05	0.90	3131	43	63	39.06	1	39	1	5
Cps_NJ1	1,161,434	1	991	43.49	0.90	3253	43	60	38.96	1	39	1	5
Cps_VS225	1,157,385	1	1054	44.11	0.90	2074	32	59	39.02	1	39	1	5
Cps_WS-RT-E30	1,140,789	1	998	43.58	0.90	3104	39	59	39.03	1	39	1	5
Csu_1-25a	1,088,751	3	902	39.80	0.88	3363	31	47	42.07	2	37	1	6
Ctr_434-Bu	1,038,842	1	891	39.39	0.90	1787	43	55	41.33	2	37	1	6
Ctr_A-HAR-13	1,044,459	1	903	39.86	0.90	1787	46	56	41.30	2	37	1	5
Ctr_D-UW-3-CX	1,042,519	1	892	39.68	0.90	1787	46	55	41.31	2	37	1	5

^a^ Plasmids not included; Abbreviations: Cab *Chlamydia (C.) abortus*, Cav *C. avium*, Cca *C. caviae*, Cfe *C. felis*, Cga *C. gallinacea*, Cib (*Cand*.) C. ibidis, Cmu *C. muridarum*, Cpe *C. pecorum*, Cpn *C. pneumoniae*, Cps *C. psittaci*, Csu *C. suis*, Ctr *C. trachomatis*.

**Table 2 pathogens-09-00899-t002:** Nucleotide identity values of PZ sequences of representative strains of all *Chlamydia* spp.

	Cab_16DC122	Cab_S26-3	Cav_10DC88	Cga_08-1274-3	Cca_GPIC	Cfe_Fe-C56	Cib_10-1398-6	Cps_6BC	Cpe_E58	Cpn_TW-183	Cmu_Nigg	Csu_1-25a	Ctr_D-UW-3-CX
Cab_16DC122		64.71	47.94	30.15	35.61	35.78	32.00	65.12	28.32	35.41	29.39	29.56	29.68
Cab_S26-3	64.71		47.01	24.48	46.87	44.55	36.17	81.55	33.44	40.42	34.56	34.99	35.33
Cav_10DC88	47.94	47.01		0.00	46.38	45.67	43.29	48.04	42.16	44.86	43.78	44.15	43.71
Cga_08-1274-3	30.15	24.48	0.00		31.10	45.30	52.16	29.73	27.58	0.00	28.40	28.07	28.58
Cca_GPIC	35.61	46.87	46.38	31.10		58.50	35.12	35.57	30.01	39.36	28.12	28.46	28.45
Cfe_Fe-C56	35.78	44.55	45.67	45.30	58.50		43.39	36.64	34.13	39.95	31.80	31.97	32.11
Cib_10-1398-6	32.00	36.17	43.29	52.16	35.12	43.39		32.88	29.61	36.88	31.33	31.22	31.64
Cps_6BC	65.12	81.55	48.04	29.73	35.57	36.64	32.88		30.07	40.63	31.15	31.30	31.51
Cpe_E58	28.32	33.44	42.16	27.58	30.01	34.13	29.61	30.07		39.82	30.16	30.33	30.23
Cpn_TW-183	35.41	40.42	44.86	0.00	39.36	39.95	36.88	40.63	39.82		39.15	38.83	38.55
Cmu_Nigg	29.39	34.56	43.78	28.40	28.12	31.80	31.33	31.15	30.16	39.15		66.24	62.96
Csu_1-25a	29.56	34.99	44.15	28.07	28.46	31.97	31.22	31.30	30.33	38.83	66.24		62.41
Ctr_D-UW-3-CX	29.68	35.33	43.71	28.58	28.45	32.11	31.64	31.51	30.23	38.55	62.96	62.41	

**Table 3 pathogens-09-00899-t003:** Plasticity zone parameters of representative strains of all *Chlamydia* spp.

Species_Strain	PZ Total Size [nt]	# CDS in PZ	Biotin Modi-Fication	*toxB* [nt]	MAC/ Perforin ^a^ [nt]	Trp Operon	Purine Synthesis and Recycling
Cab_16DC122	22,240	13	*accB, accC*	9312	-	-	-
Cab_S26-3	11,776	14	*accB, accC*	-	(681)	-	*guaB_1/2* ^b^
Cav_10DC88	5694	6	*accB, accC*	-	-	-	-
Cca_GPIC	34,753	21	*accB, accC*	10,041	(504) ^b^	*trpA, trpB_1/2, trpD, trpF, trpR, kynU*	*guaA, guaB*, ADA
Cfe_Fe-C56	39,924	28	*accB, accC*	9897	2442; (501)	*trpA, trpB, trpD, trpF, trpR, kynU*	*guaA, guaB*, ADA
Cga_08-1274-3	15,845	8	*accB, accC*	9363	-	-	-
Cib_10-1398-6	31,344	20	*accB, accC*	9378	2505; 2484	-	-
Cmu_Nigg	82,115	45	*accB, accC*	9657; 10,008; 9768	2430	-	*guaA, guaB*, ADA
Cpe_E58	42,163	18	*accB, accC*	10,134; 10,317	2418	-^c^	*guaA, guaB*, ADA
Cpn_TW-183	8759	11	*accB, accC*	-	1236	-	*guaB_1/2* ^b^
Cps_6BC	29,145	16	*accB, accC*	10,074	2469; (627)	-	*guaA, guaB*, ADA
Csu_1-25a	82,505	52	*accB, accC*	10,089; 9675	2433	*trpA, trpB, trpR*	-
Ctr_D-UW-3-CX	55,445	49	*accB, accC*	-	2433	*trpA, trpB, trpR*	-

^a^ in parenthesis: size of MAC/perforin domain-containing protein; ^b^ pseudogenes; ^c^
*C. pecorum* strain E58 has Trp operon outside PZ (ref. 1); *accB* Biotin carboxyl carrier protein of acetyl-CoA carboxylase, *accC* Biotin carboxylase; *guaA* GMP synthase, *guaB* inosine-5’-monophosphate dehydrogenase, ADA adenosine/AMP deaminase.

**Table 4 pathogens-09-00899-t004:** Members of the Pmp family identified in *Chlamydia* spp.

Strain	# pmp	Individual Pmps in Subtypes					
		A	B	D	E	G	H
Cab_S26-3	18	2	1	18	3,4,5	7, 8, 9, 10, 11, 12, 13, 14, 15, 16, 17	6
Cav_10DC88	7	19	B	21	15	G, 13 or G-I	14
Cca_GPIC	18	A	B	D	E/F [5]	G [9]	H
Cfe_Fe-C56	20	19	20	1	14, 15, 16, 17, 18	2, 3, 4, 5, 6, 7, 8, 9, 10, 11, 12	13
Cga_08-1274-3	10	A	B	D	E, F	G/I [4]	H
Cib_10-1398-6	22	2	1	22	3, 4, 5	8, 9, 10, 11, 12, 13, 14, 15, 16, 17, 18, 19, 20, 21	6, 7
Cpe_E58	15	A	B	D	E [2]	G [9]	H
Cpn_TW-183	21	19	20	21	15, 16, 17, 18	1, 2, 3, 4, 5, 6, 7, 8, 9, 10, 11, 12, 13	14
Cps_6BC	21	2	1	22	3, 4, 5	7, 8, 9, 10, 11, 12, 13, 14, 15, 16, 17, 19, 20, 21	6
Cmu_Nigg	9	A	B, C	D	E, F	G, I	H
Csu_1-25a	9	A	B, C	D	E, F	G, I	H
Ctr_D-UW-3-CX	9	A	B, C	D	E, F	G, I	H

[in square brackets]: no. of subtype members.

**Table 5 pathogens-09-00899-t005:** Characteristic genomic features of *Chlamydia* spp. with different host preference.

Strain	PZ Size [nt]	ToxB [aa]	Trp Operon *	# Pmps	Inc Family Subtypes	SINC [aa]	HctA/HctB [aa]	Main Host
Cab_S26-3	11,776			18	A, B, C, V, X, Y	361	123/154	ruminant
Cps_6BC	29,145	3358		21	A, B, C, V, X, Y	502	116-123/197	avian
Cav_10DC88	5694			7	A, B, C, V	234	-/86 **	avian
Cga_08-1274-3	15,845	3121		10	A, B, C, V	237	-/187	avian
Cib_10-1398-6	31,344	3126		22	A, B, C, V	242	118/199	avian
Cca_GPIC	34,753	3347	7	18	A, B, C, V	238	125/152	rodent
Cfe_Fe-C56	39,924	3299	6	20	A, B, C, V	240	126/168	feline
Cpe_E58	42,163	3378; 3439	***	21	A, B, C		118/190	ruminant
Cpn_TW-183	8759			21	B, C		123/172	human
Cmu_Nigg	82,115	3219; 3336; 3256		15	A, B, X		125/207	rodent
Csu_1-25a	82,505	3363; 3225	3	9	A, B, C, D, E, F, G		126/203	porcine
Ctr_D-UW-3-CX	55,445		3	9	A, B, C, D, E, F, G, V		125/201	human

* no. of genes, ** truncated, X, Y unassigned to subtype (Y also designated NC), *** strain E58 has Trp operon outside PZ (ref. 1).

**Table 6 pathogens-09-00899-t006:** Characteristics of the strains included in this study.

Species	Strain ^1^	Source	NCBI acc. no.	ENA	*de novo* seq. (Source)
*Chlamydia abortus*	Cab_16DC122	Muscovy duck			FLI ^2^
	Cab_C18-98 (B577^T^)	Sheep	SAMEA1094359		
	Cab_S26-3	Sheep	NC_004552.2		
*Chlamydia avium*	Cav_10DC88^T^	Pigeon	NZ_CP006571.1	GCA_000583875.1 ASM58387v1	
	Cav_11DC096	Pigeon			FLI ^2^
	Cav_12DC097	Pigeon			FLI ^2^
	Cav_14DC103	Pigeon			FLI ^2^
*Chlamydia caviae*	Cca_GPIC^T^	Guinea pig	NC_003361.3		
*Chlamydia felis*	Cfe_Fe-C56^T^	Cat	NC_007899.1		
*Chlamydia gallinacea*	Cga_08-1274-3^T^	Chicken	NZ_CP015840.1		
	Cga_12-4358	Chicken			ANSES ^3^
4 contigs	Cga_13DC099	Turkey			FLI ^2^
3 contigs	Cga_14DC100	Chicken			FLI ^2^
3 contigs	Cga_14DC101	Chicken			FLI ^2^
	Cga_JX-1	Chicken	NZ_CP019792.1. CP019792.1	GCA_002007725.1. ASM200772v1	
*Cand.* Chlamydia ibidis	Cib_10-1398-6	Ibis	NZ_APJW01000001.1		
*Chlamydia muridarum*	Cmu_Nigg^T^	Mouse	NC_002620.2		
*Chlamydia pecorum*	Cpe_E58^T^	Cattle	NC_015408.1		
*Chlamydia pneumoniae*	Cpn_TW-183^T^	Human	NC_005043.1		
*Chlamydia psittaci*	Cps_02DC15	Cattle	NC_017292.1	GCA_000415545.1	
	Cps_08DC60	Human	NC_017290.1	GCA_000270445.1	
	Cps_6BC^T^	Parakeet	NC_015470.1	GCA_000191925.1	
	Cps_CP3	Pigeon	NC_018625.1	GCA_000298535.2	
	Cps_GR9	Duck	NC_018620.1	GCA_000298415.1	
	Cps_Mat116	Psittacine	CP002744.1		
	Cps_MN	Human	NC_018627.1	GCA_000298435.2	
	Cps_NJ1	Turkey	CP003798.1		
	Cps_VS225	Psittacine	CP003793.1		
	Cps_WS-RT-E30	Duck	NC_018622.1	GCA_000298475.2	
*Chlamydia suis*	Csu_1-25a	Swine	FTQL01000001		
*Chlamydia trachomatis*	Ctr_434-Bu	Human	NC_010287.1		
	Ctr_A-HAR-13^T^	Human	NC_007429.1		
	Ctr_D-UW-3-CX	Human	NC_000117.1		

^1^ superscript T = type strain. ^2^ Chlamydia & Mycoplasma Group, Inst. Molec. Pathogenesis, Friedrich-Loeffler-Institut, Jena, Germany. ^3^ ANSES, Bacterial Zoonoses Unit, Maisons-Alfort, France.

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
