# Peer review of "Comparative Genome Analysis of 33 Chlamydia Strains Reveals Characteristic Features of Chlamydia Psittaci and Closely Related Species"

_pathogens, 2020, doi:10.3390/pathogens9110899_

Round 1

Reviewer 1 Report

This manuscript describes a study that indentified genome-based features of avian and human Chlamydia psittaci and related chlamydiae. Thirty-three whole genome sequences were analysed here

Comments

This study includes a large collection of chlamydial strains with special emphasis on the previously quite neglected ones (ie. C. psittaci and related strains). They also use novel methods Roary ILP (currently under development) that are freely available in analyses and produce novel data concerning the strains.

This seems to be a well-planned and well-executed study. My suggestion is to condense the introduction (the text is longish, and some paragraphs are quite short ).

Reviewer 2 Report

No underlying hypothesis is mentioned anywhere in the text, but the author’s appear to want to ascertain whether the inclusion of ‘hypothetical genes’ in the construction of their putative ‘core genome’ has a significant impact on calculated relatedness between strains / species.

The authors do a good job of what they set out to do: analyze whole-genome sequences of 33 strains belonging to 12 species. Not surprisingly, they found commonalities and difference at both the species and genus level. This data will no doubt contribute to the ongoing discussion about relatedness within the Chlamyidae. Their use of all this sequencing data to generate a Phylogenetic reconstruction based on the core genome predicted by RIBAP for all 33 genomes will be of value to the field, particularly as they include hypothetical genes in their analysis. The author's mention this in their discussion, and should expand on this point, as it is the most important contribution to the field from this study. 

What the authors do not do a good job of is directly addressing why these similarities / differences are (or are not) important, preferring to leave functional studies to future researchers. More emphasis should be placed on the conservation (or not) of core metabolic genes if tissue tropism is to be adequately addressed. They make some general comments and assertions within the discussion section, but it would have been nice to more directly address one or two of these hypotheses in the manuscript itself. As an example, the presence of purine nucleotide biosynthesis / salvage genes in the plasticity zone appears to be a major delineator of speciation, yet the author’s do not propose how this is likely to impact Chlamydia species biology or whether it directly represents genes essential for certain infectious cites, a likely explanation for highly prevalent tissue tropisms. Details on purine availability within the hosts / specific tissue cites of infection would allow for this hypothesis to be weighed against the known literature, since the authors do not wish to address the hypothesis directly in their study. 

Additionally, it is unclear whether the authors have sufficiently accounted for genes known to be non-functional in some species, but that do not appear to have common genetic decay markers such as early stop codons or frameshift mutations. Most of this work has been conducted in C. trachomatis utilizing E.coli surrogate systems, with the arginine decarboxylase system ArgDC / AaxABC as a good example. These nonfunctional genes / pseudogenes are likely to be found in more recently divergent organisms, prior to more recognizable genetic decay markers such as frame shifts / deletions. A more detailed analysis of known pseudogenes in from C. trachomatis in the other species would be informative and also contribute to the underlying theme of host / tissue tropism.
